**Data Availability Statement:** All relevant data are within the paper and its Supporting Information files.

**Funding:** This work was supported by JSPS KAKENHI Grant Number JP20H03592. The funders

# Anemia in female patients with myasthenia gravis

**Koji Sekiguchi**[1], **Kei Ishizuchi**[1], **Tsubasa Takizawa**[1], **Haruhiko Motegi**[1,2], **Munenori Oyama**[1], **Jin Nakahara**[1], **Shigeaki Suzuki**[1] *

1 Department of Neurology, Keio University School of Medicine, Tokyo, Japan, 2 Department of Neurology, The Jikei University School of Medicine, Tokyo, Japan

* sgsuzuki@keio.jp

## Abstract

Myasthenia gravis (MG) is the most common autoimmune neuromuscular disorder, and is more common in women than in men. Anemia is also more common in women. The purpose of this study was to investigate factors associated with anemia and the negative impact of anemia in female MG patients. We investigated factors related to MG and anemia in 215 female patients with MG, who were attending the MG clinic of Keio Hospital between January and December 2021. We statistically evaluated clinical factors related to anemia in patients with and without anemia. Eighty-five patients (40%) had anemia in the past, and 130 patients did not have anemia in the past. There were no significant differences in age at study, age at MG onset, body mass index, or frequency of autoantibodies between the anemia and non-anemia groups. MG severity evaluated by the MG Foundation of America classification was greater in the anemia group than in the non-anemia group. History of anemia was associated with immunosuppressive treatment, such as prednisolone and calcineurin inhibitor treatment. There was a correlation between hemoglobin levels and the MG-quality of life score. Long term immunosuppressive therapy can cause anemia in female MG patients. Anemia may negatively affect the quality of life of female MG patients.

## Introduction

Myasthenia gravis (MG) is the most common autoimmune neuromuscular disorder and is mediated by autoantibodies to acetylcholine receptors or muscle-specific tyrosine kinase [1]. It affects a variety of ages and affects women more predominantly than men. Early-onset MG, which is defined as MG with onset under age 50, is three times as likely to be diagnosed in females as in males, whereas males slightly outnumber females in the late-onset group [2]. Recent advances in immunosuppressive treatment have dramatically reduced the mortality of MG, but MG patients still find it difficult to maintain their activities of daily living (ADL) and quality of life (QOL) due to the long-term side effects [3]. In addition, special consideration is necessary for the management of female MG patients, because they may experience pregnancy, delivery, menopause, and gynecological disorders.

Anemia is the most frequent derailment of physiology in the world throughout the life of a woman. Anemia is one of the world's leading causes of disability and thus one of the most

had no role in study design, data collection and analysis, decision to publish, or preparation of the manuscript.

**Competing interests:** The authors have declared that no competing interests exist.

serious global public health issues. From 1999 to 2020, anemia affected 33% of the worldwide population, or about 40% of women [4]. Common causes of anemia are poor nutrition, iron deficiencies, micronutrients deficiencies including deficiencies in folic acid and vitamins, infectious diseases and genetically inherited hemoglobinopathies such as thalassemia. In addition, autoimmune-mediated disorders including hemolytic anemia, paroxysmal nocturnal hemoglobinuria, and pure red cell aplasia can also cause anemia. The causes of anemia are complex and diverse, and some of them are drug-related. However, the prevalence and causes of anemia and impact of anemia on the QOL of female MG patients have not been elucidated. In addition, it is often difficult to distinguish between anemia and MG in clinical practice, in part because respiratory distress is a common symptom of both.

The purpose of the present study was to examine the cause of anemia and the negative impact of anemia in female MG patients.

## Patients and methods

Data on 388 MG patients who were followed at the MG clinic of Keio Hospital between January and December 2021 were included in this single-center retrospective analysis. The diagnosis of MG was based on clinical findings (fluctuating symptoms with easy fatigability and recovery after rest) along with the amelioration of symptoms after an intravenous administration of acetylcholinesterase inhibitors, decremental muscle response to a train of low-frequency repetitive nerve stimuli, or the presence of autoantibodies [1]. Clinical status and severity were determined by the recommendations of the Myasthenia Gravis Foundation of America (MGFA) [5]. Current MG status was evaluated by three representative clinical scales: MG-ADL, MG composite, and the revised scale of the 15-item MG-QOL [3, 6, 7]. The quantitative (QMG) scores were evaluated in the worst condition during the entire courses. In contrast, the MG-ADL, MG composite, and the revised scale of the 15-item MG-QOL were recorded in the last outpatient visit. Due to COVID-19 pandemic, we could not easily perform pulmonary function tests. Thus, we utilized MG composite instead of QMG score.

We evaluated chronological records of hemoglobin levels after the diagnosis of MG. Hemoglobin levels less than 11 g/dL in non-pregnant women aged 15 years or older were defined as moderate or severe anemia [8]. Causes and underlying disorders for developing anemia were retrospectively investigated using their medical records.

All clinical information was collected after the patients provided written informed consent. All study protocols were specifically approved by the institutional review board of Keio University (#20090278). These clinical investigations were conducted according to the principles expressed in the Declaration of Helsinki.

Comparisons between two groups were made using the unpaired $t$-test, and the chi-square test. We used a simple linear regression to estimate the relationship between hemoglobin levels and the MG-ADL, MG composite, and revised scale of the 15-item MG-QOL. The Kruskal-Wallis test was used to compare hemoglobin levels among the three groups divided by respiratory score. The statistical analyses were performed using GraphPad Prism version 9 statistical software (San Francisco, La Jolla, CA).

## Results

### Clinical features of MG patients with anemia

We followed 388 MG patients between January 2021 and December 2021, among whom 254 female MG patients were included in the present study (Fig 1). We then excluded 9 female MG patients who had transient anemia due to pregnancy, childbirth, or surgery, and 30 female patients in whom the clinical scores of MG had not been examined. A total of 215 female MG

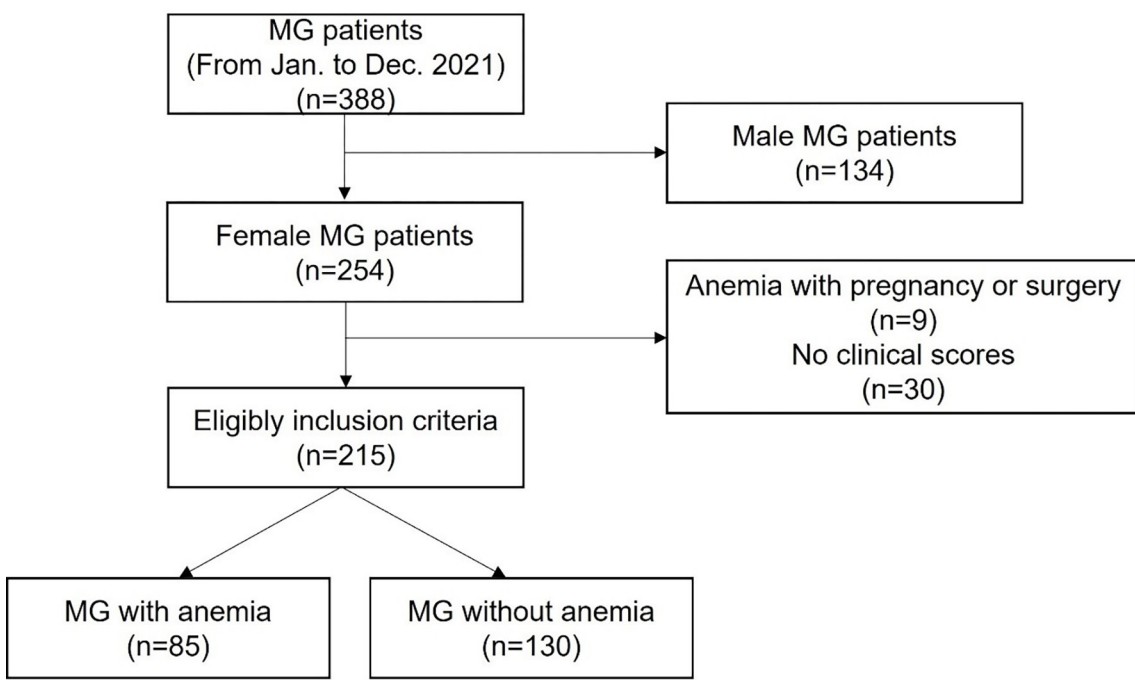

**Fig 1. Study flow.** MG, myasthenia gravis.

patients were thus included in the present study. A total of 85 (40%) of the 215 patients experienced an episode of moderate or severe anemia (hemoglobin levels less than 11 g/dL) after the diagnosis of MG.

To elucidate the clinical features of female MG patients with anemia, we divided the 215 patients into an anemia group (n = 85) and a non-anemia group (n = 130) (Table 1). There were no differences in age, disease duration of MG, body mass index, frequency of smoking habits, or menopause between the two groups. However, MG severity evaluated by MGFA classification was greater in the anemia group than in the non-anemia group ($p$ = 0.003, Fig 2A). Bulbar involvement, but not myasthenic crisis was significantly more common in the anemia group compared to the non-anemia group. QMG scores under the worst condition during the entire course were greater in the anemia group compared to the non-anemia group (15.8 ± 6.2 versus 13.1 ± 5.8, $p$ = 0.002). There were no significant differences in the serological profiles between the two groups.

Next, treatment regimens of MG were compared between the anemia and non-anemia groups. All immunotherapies, including prednisolone, calcineurin inhibitors, plasma pheresis, intravenous immunoglobulin, and anti-complement, were more frequently used in the anemia group compared to the non-anemia group. Female MG patients in the anemia group frequently underwent thymectomy, although there was no significant difference in the percentage of patients with thymoma between the anemia and non-anemia groups. There was also no significant difference in the MGFA post-intervention status between the anemia and non-anemia groups ($p$ = 0.051, Fig 2B).

## Causes of anemia

Laboratory findings revealed that 49 (23%) of the 215 female MG patients had iron-deficiency anemia. To elucidate the disorders causing the anemia, we investigated the frequencies of gynecological disorders and gastrointestinal disorders that occurred after the diagnosis of MG.

**Table 1. Comparison between female MG patients with and without moderate or severe anemia.**

| Characteristics | Anemia | Non-anemia | p-value |
|---|---|---|---|
| n (%) | (n = 85) | (n = 130) | |
| Age, mean ± SD | 54.2 ± 16.8 | 56.1 ± 16.7 | 0.42 |
| Disease duration, years, mean ± SD | 13.8 ± 11.7 | 13.0 ± 17.7 | 0.71 |
| Body mass index (kg/m$^2$), mean ± SD | 21.3 ± 4.4 | 21.4 ± 3.7 | 0.84 |
| Smoking | 23 (27) | 22 (17) | 0.074 |
| Menopause | 46 (54) | 83 (64) | 0.15 |
| Bulbar involvement | 52 (61) | 58 (45) | 0.018* |
| Crisis | 10 (12) | 6 (5) | 0.051 |
| Thymoma | 16 (19) | 19 (15) | 0.45 |
| Quantitative MG score, mean ± SD | 15.8 ± 6.2 | 13.1 ± 5.8 | 0.002* |
| Anti-acetylcholine receptor positive | 58 (68) | 77 (59) | 0.18 |
| Anti-muscle-specific tyrosine kinase positive | 5 (6) | 8 (6) | 0.93 |
| Treatment | | | |
| Prednisolone | 68 (80) | 79 (61) | 0.003* |
| Calcineurin inhibitors | 49 (58) | 50 (39) | 0.006* |
| Plasmapheresis | 22 (26) | 11 (8) | 0.0005* |
| Intravenous immunoglobulin | 39 (46) | 27 (21) | <0.0001* |
| Anti-complement | 4 (5) | 0 (0) | 0.013* |
| Thymectomy | 40 (47) | 33 (25) | 0.001* |
| Underlying disorders | | | |
| Gynecological disorders | 31 (37) | 20 (15) | 0.0004* |
| Gastrointestinal disorders | 18 (21) | 6 (5) | 0.0002* |

MG: myasthenia gravis; SD: standard deviation.

*Statistically significant.

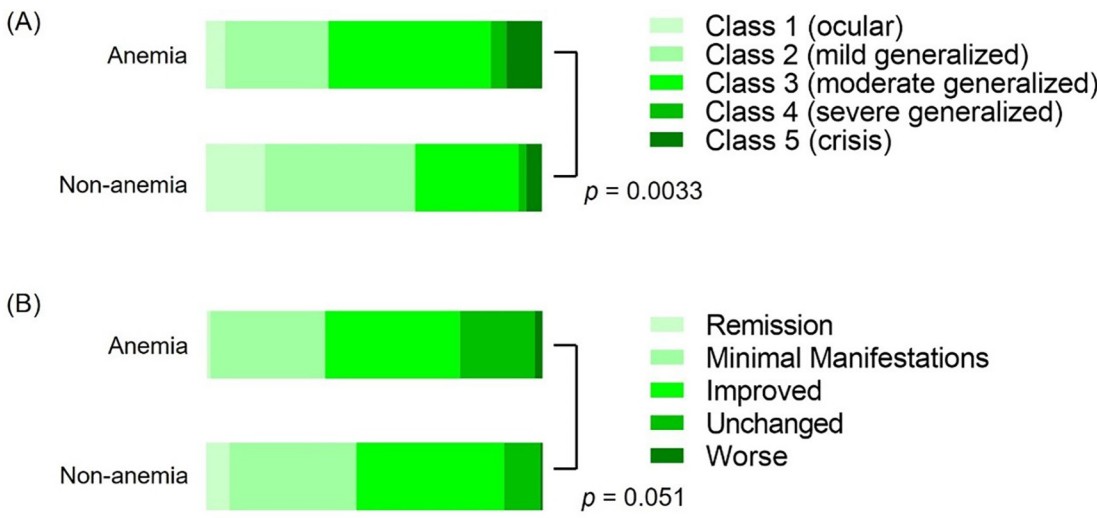

**Fig 2.** Disease classification (A) and post-interventional status (B) of 215 female myasthenia gravis (MG) patients. These assessments followed the recommendations of the MG Foundation of America.

We found that female MG patients in the anemia group suffered from these disorders more frequently than those in the non-anemia group. Among 49 MG patients with iron deficiency anemia, 35 (71%) had hemoglobin levels within the normal range at the time of the latest hemoglobin count by iron supplementation.

We further evaluated the causes of the underlying anemia based on the detailed clinical courses of the female MG patients with anemia. Chronic infections arising from intra-abdominal abscess, aspiration pneumonia, and infective endocarditis were causative of anemia in 3 (1%) patients. None had parasite infection such as hookworm and schistosomiasis. There were no patients with thalassemia. Regarding concomitant autoimmune diseases, 18 (21%) of the 85 patients suffered from other diseases (S1 Table). Although some of these autoimmune diseases carry a risk of anemia, our analysis indicated that none of the diseases were responsible for the anemia observed in our patients, with the exception of a thymoma-associated patient with pure red cell aplasia.

As far as information on the lifestyle of the 85 female MG patients, it was likely that insufficient intake of resulting from poverty, malnutrition and diet was not directly the causative factor of anemia. Causes and underlying diseases were not identified in 27 (32%) of the 85 MG patients. There was no clear difference in the latest hemoglobin levels between the anemia group with unknown cause and the anemia group with cause (12.5±1.7, 12.2±1.9, $p = 0.46$). We thus considered that the anemia in these patients may have been due to long-term immunotherapy.

## Correlation between hemoglobin and MG status

Treatment of the underlying diseases and iron supplementation were effective in 64 (75%) of 85 female MG patients with anemia; the hemoglobin levels in these 64 patients exceeded 11 g/dL in the final examination. On the other hand, in the remaining 21 (25%) patients the hemoglobin levels were still below 11 g/dL.

Next, we studied the negative impact of anemia on the MG status using the final hemoglobin levels in 215 female MG patients with and without anemia. The correlation between the final hemoglobin levels and three representative scales—i.e., the MG-ADL, MG composite, and the revised scale of the 15-item MG-QOL—was evaluated (Fig 3). There was no correlation between hemoglobin levels and the MG-ADL or MG composite scales. However, there was a statistically weak correlation between the revised scale of the 15-item MG-QOL score and hemoglobin levels ($r = 0.16$ and $p = 0.02$). The analysis thus indicated that decreased hemoglobin levels may be associated with QOL impairment of female MG patients.

Finally, we evaluated the association between breathing conditions and hemoglobin levels in the 215 female MG patients with anemia. For this purpose, we divided patients into three groups according to their respiratory symptoms from the MG-ADL score: no symptoms (0, n = 121), dyspnea on exertion (1, n = 73), and dyspnea at rest and respiratory support (2 and 3, n = 21). The median hemoglobin concentrations of these three groups were 13.3 mg/dL, 13.0 mg/dL, and 13.2 mg/dL (Fig 4), respectively. There was no association between breathing condition stratified by MG-ADL scale and hemoglobin levels.

## Discussion

The results of this single-center retrospective study can be summarized as follows: (i) 85 (40%) of our female MG patients had moderate or severe anemia after the diagnosis of MG, (ii) female MG patients with anemia presented with severe muscle weakness, which resulted in their receiving more intensive immunotherapy, (iii) 27 (32%) patients had no known cause of anemia, and (iv) hemoglobin levels were associated with the revised scale of the 15-item MG-QOL, but not with the MG-ADL or MG composite scores.

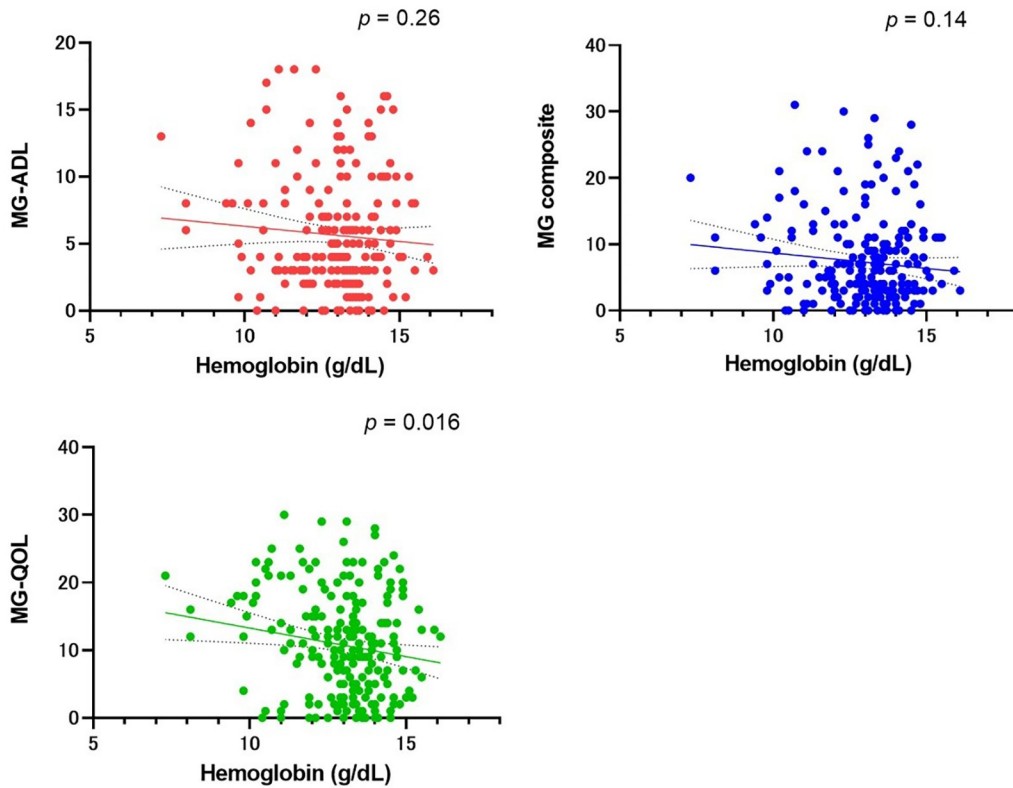

**Fig 3. Correlation between hemoglobin levels and myasthenia gravis (MG) status of 215 female MG patients.** MG status was evaluated by the MG activities of daily living (MG-ADL), MG composite, and the revised scale of the 15-item MG quality of life (MG-QOL). Note that these scores are higher in patients with more severe MG.

MG is an organ-specific autoimmune disorder and is not accompanied by anemia, in contrast to systemic autoimmune disorders such as systemic lupus erythematosus [1]. However, the present study indicated that anemia constituted a significant disadvantage for female MG patients in the clinical settings. Although female MG patients with anemia suffered from severe muscle weakness, their post-intervention status was improved, as observed in those without anemia. We attributed this successful disease control to the intensive immunotherapy performed in these patients, which was usually longer than 10 years. On the other hand, long-term immunotherapy for MG also caused various side effects, including anemia.

Many reports indicate that iron-deficiency remains the top cause of anemia [9]. In fact, well-known causes of iron-deficiency anemia, such as gynecological diseases and gastrointestinal diseases, were more frequently found in our MG patients with anemia. In addition, we think that female MG patients receiving intensive and long-term immunotherapy may be vulnerable to iron metabolism [9]. It has also been noted that proton-pump inhibitors and histamine-2 receptor antagonists that are often used in conjunction with prednisolone are a frequently overlooked cause of impaired iron absorption [10]. The simultaneous occurrence of multiple causes of iron-deficiency anemia is thus common, and regular blood sampling is required for best management of female MG patients.

Various additional autoimmune diseases are observed in 10%-15% of MG patients [11]. Pure red cell aplasia is occasionally observed in patients with thymoma-associated MG. We reported that 3 (4%) of 72 female MG patients developed pure red cell aplasia in a previous cohort of the Keio Hospital between 1991 and 2000 [12]. These patients developed pure red

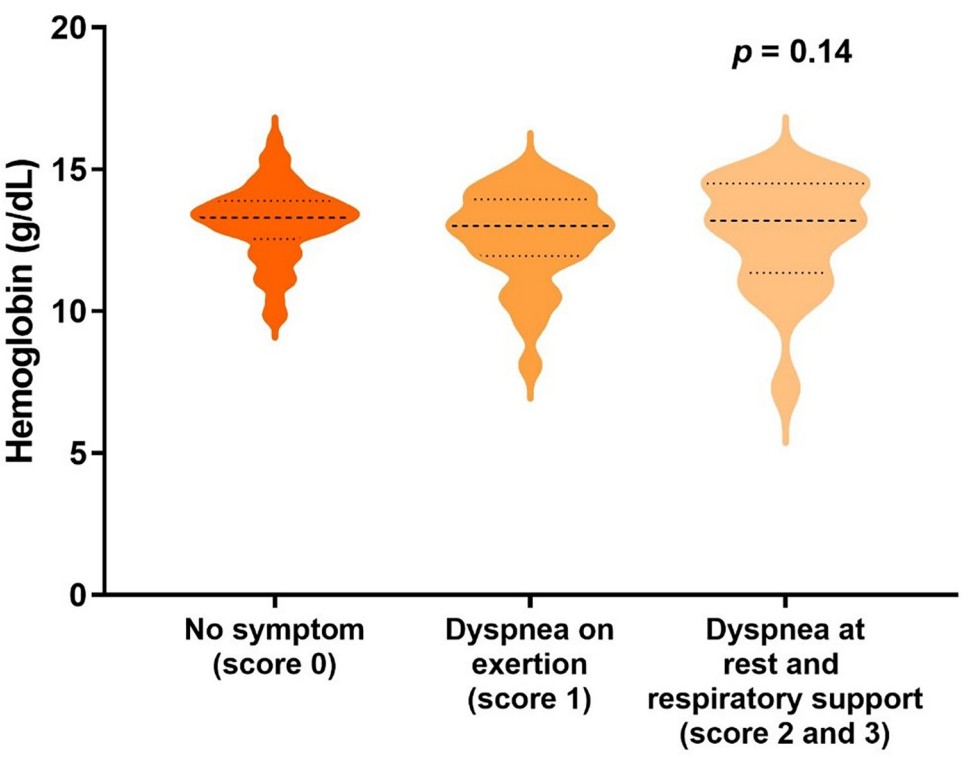

**Fig 4. Correlation between breathing status stratified by myasthenia gravis activities of daily living (MG-ADL) and hemoglobin levels in 215 female MG patients with anemia.** Dotted lines indicate quartile levels.

cell aplasia 2–19 years after the removal of thymoma, during a period when MG was well-controlled. In the present study, we observed a 29-year-old woman with thymoma-associated MG in whom cyclosporine was effective in both MG and pure red cell aplasia.

The strength of the present study is that we evaluated the association between hemoglobin levels and three clinical scales at the same time. Hemoglobin levels were not correlated with the severity of MG as evaluated by the MG-ADL and MG composite scales. However, we demonstrated that decreased levels of hemoglobin were associated with QOL impairment of female MG patients. We consider that this statistical difference in the MG-QOL score, which is a disease-specific measure, warrants further investigation. On the other hand, one of the limitations of this study is that we were not able to evaluate whether the MG-QOL scores improved before or after anemia was improved. Whether correction for anemia improves MG-QOL scores is thus a subject for future research. Another limitation is that anemia in female MG patients may be due to long-term use of immunosuppressive drugs, but the use of immunosuppressive drugs is also a confounding factor with the severity of MG. When the groups were divided by the presence or absence of anemia in the latest hemoglobin level, the worst QMG score ever was not statistically significant. On the other hand, the use of calcineurin inhibitors was statistically significantly higher in the anemic group (S2 Table).

Weakness, fatigue, dyspnea on exertion, difficulty in concentration, and poor work productivity are common symptoms of both anemia and MG. Among them, the assessment of dyspnea is especially challenging. When MG patients with anemia complain of dyspnea, it is

difficult to determine whether the dyspnea is due to MG or anemia. To resolve this question, we examined the relationship between respiratory condition and hemoglobin levels. However, we failed to detect a correlation between hemoglobin levels and the respiratory-related items of the MG-ADL scale [6]. In this context, our inability to perform pulmonary function tests in MG patients due to the impact of the COVID-19 pandemic on our hospital constitutes a study limitation.

## Conclusions

With modern immunosuppressive, symptomatic, and supportive treatments, the prognosis for MG patients is generally favorable. However, long-term drug treatment is usually necessary for most MG patients. In addition to various adverse events of corticosteroids and non-steroidal immunosuppressants, female MG patients suffer from anemia. Anemia which is suspected to be related to long-term immunosuppressive drugs, causes the impairment of patients' QOL. Monitoring hemoglobin levels is necessary for best management of female MG patients.

## Supporting information

**S1 Data.**
(XLSX)

**S1 Table. Complications of autoimmune diseases.** * One patient has Grave's disease and neuromyelitis optica spectrum disorders, and one patient has Grave's disease and stiff-person syndrome, one patient has Systemic lupus erythematosus and Sjögren's syndrome. ** One patient has Hashimoto's disease and type 1 diabetes mellitus.
(DOCX)

**S2 Table. Comparison between female MG patients with and without moderate or severe anemia at latest hemoglobin level.** CNIs: calcineurin inhibitors, QMG score: quantitative myasthenia gravis score; SD: standard deviation. *: $p < 0.05$, **: $p < 0.01$.
(DOCX)

## Author Contributions

**Data curation:** Kei Ishizuchi.

**Formal analysis:** Koji Sekiguchi.

**Investigation:** Shigeaki Suzuki.

**Writing – original draft:** Koji Sekiguchi.

**Writing – review & editing:** Kei Ishizuchi, Tsubasa Takizawa, Haruhiko Motegi, Munenori Oyama, Jin Nakahara, Shigeaki Suzuki.

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
