## [Decision Letter · Decision Letter 0]

27 May 2022

PONE-D-22-12752Anemia in female patients with myasthenia gravisPLOS ONE

Dear Dr. Suzuki,

Thank you for submitting your manuscript to PLOS ONE. After careful consideration, we feel that it has merit but does not fully meet PLOS ONE’s publication criteria as it currently stands. Therefore, we invite you to submit a revised version of the manuscript that addresses the points raised during the review process. The major concern of this article is there are several confounding factors when analyzing the association between anemia and MG severity and quality of life. Please adjust them in the revised manuscript.

We look forward to receiving your revised manuscript.

Kind regards,

Jong-Ling Fuh

Academic Editor

PLOS ONE

Journal Requirements:

2.Thank you for stating the following financial disclosure: 

 "This work was supported by JSPS KAKENHI Grant Number JP20H03592.". 

 "None". 

Reviewers' comments:

Reviewer's Responses to Questions

**Comments to the Author**

1. Is the manuscript technically sound, and do the data support the conclusions?

Reviewer #1: Yes

Reviewer #2: Yes

Reviewer #3: No

2. Has the statistical analysis been performed appropriately and rigorously? 

Reviewer #1: Yes

Reviewer #2: Yes

Reviewer #3: No

3. Have the authors made all data underlying the findings in their manuscript fully available?

Reviewer #1: Yes

Reviewer #2: Yes

Reviewer #3: Yes

4. Is the manuscript presented in an intelligible fashion and written in standard English?

Reviewer #1: Yes

Reviewer #2: Yes

Reviewer #3: No

5. Review Comments to the Author

Reviewer #1: The authors report 40% of 215 female patients with myasthenia gravis (MG) had anemia in the past and long-term immunosuppressive therapy might cause anemia with a negative impact on the quality of life (QoL) of female MG patients.

The paper was well-written and might be accepted for publish with minor revision.

MAJOR CRITICISM

Iron deficiency anemia (IDA) was the most common cause of anemia in this cohort (49/85=58%) and controlled by iron supplementation. The underlying causes of IDA should be provided and the response to iron supplementation should be analyzed also. The second most common cause of anemia was presumed to be drug-related in 32% of 85 MG patients based on lack of possible underlying disease contributory to anemia. Was this subgroup of anemic patient more resistant to anemia treatment? Subgroup analysis of anemia with all MG outcome measures might provide more clues for the true impact of MG drugs on QoL.

MINOR CRITICISM

The total number of MG cohort seems to be inconsistent. 383 patients in page 4 (Patients and methods) but 385 patients in page 5 (Results).

Were the quantitative assessments including QMG, MG-ADL, and MG-QoL performed on routine basis or at the last outpatient visit?

Reviewer #2: The purpose of this study was to investigate factors associated with anemia and the negative impact of anemia in female MG patients. The authors investigated factors related to MG and anemia in 215 female patients with MG.

According to their results 40% of the female MG patients had moderate or severe anemia after the diagnosis of MG, female MG patients with anemia presented with severe muscle weakness, which resulted in their receiving more intensive immunotherapy, 32% of the patients had no known cause of anemia, and hemoglobin levels were associated with the revised scale of the 15-item MG-QOL.

They conclude that long-term immunotherapy for MG may cause various side effects, including anemia in female MG patients.

It is a very interesting study on MG, since so far prevalence and causes of anemia and impact of anemia on the QOL of MG patients have not been elucidated. Important study for the clinicians, as the negative impact of anemia in MG patients may be prevented.

Reviewer #3: This is a retrospective study on the association between anemia and MG severity/quality of life in female patients. Female MG patients were divided into two groups according to the presence of anemia, and the authors found that mg severity was greater in anemia group and there was a correlation between hemoglobin level and quality of life. The authors should have considered several confounding factors when analyzing the association between anemia and MG severity and quality of life. In particular, since long-term immunosuppressive treatment is related to both anemina and mg severity, it is necessary to adjust the effect using methods such as stratification or multivariate regression.

6. PLOS authors have the option to publish the peer review history of their article (what does this mean?). If published, this will include your full peer review and any attached files.

Reviewer #1: No

Reviewer #2: **Yes: **Paraskevi Zisimopoulou

Reviewer #3: No

---

## [Author Response · Author response to Decision Letter 0]

25 Jun 2022

Thank you very much for your letter of May 28, 2022, regarding our manuscript entitled, " Anemia in female patients with myasthenia gravis". We thank the reviewers for their comments; our responses and the changes made (or reasons for not making changes) are summarised in the file 'Response to reviewers_0620'.

---

## [Decision Letter · Decision Letter 1]

2 Aug 2022

PONE-D-22-12752R1Anemia in female patients with myasthenia gravisPLOS ONE

Dear Dr. Suzuki,

Thank you for submitting your manuscript to PLOS ONE. After careful consideration, we feel that it has merit but does not fully meet PLOS ONE’s publication criteria as it currently stands. Therefore, we invite you to submit a revised version of the manuscript that addresses the points raised during the review process.

Please revise the last paragraph (Discussion section, conclusion part) with key messages that the article intends to deliver as the reviewer suggested.

We look forward to receiving your revised manuscript.

Kind regards,

Jong-Ling Fuh

Academic Editor

PLOS ONE

Journal Requirements:

Reviewers' comments:

Reviewer's Responses to Questions

**Comments to the Author**

1. If the authors have adequately addressed your comments raised in a previous round of review and you feel that this manuscript is now acceptable for publication, you may indicate that here to bypass the “Comments to the Author” section, enter your conflict of interest statement in the “Confidential to Editor” section, and submit your "Accept" recommendation.

Reviewer #1: (No Response)

Reviewer #3: All comments have been addressed

2. Is the manuscript technically sound, and do the data support the conclusions?

Reviewer #1: Yes

Reviewer #3: Yes

3. Has the statistical analysis been performed appropriately and rigorously? 

Reviewer #1: Yes

Reviewer #3: Yes

4. Have the authors made all data underlying the findings in their manuscript fully available?

Reviewer #1: Yes

Reviewer #3: Yes

5. Is the manuscript presented in an intelligible fashion and written in standard English?

Reviewer #1: Yes

Reviewer #3: Yes

6. Review Comments to the Author

Reviewer #1: All the queries were responded well. Re-analysis of anemia cohort had been done and the results were confident. There were no concern about research ethics. This manuscript was well-written and might be accepted for publication.

Reviewer #3: It would be better to supplement the last paragraph (Discussion section, conclusion part) with key messages that the article intends to deliver.

7. PLOS authors have the option to publish the peer review history of their article (what does this mean?). If published, this will include your full peer review and any attached files.

Reviewer #1: **Yes: **Jiann-Horng Yeh

Reviewer #3: **Yes: **Yoon-Ho Hong

---

## [Author Response · Author response to Decision Letter 1]

10 Aug 2022

Thank you very much for your valuable time and careful peer review. We have deleted the last sentence of the discussion and added a "Conclusions" section.

---

## [Editor Report · Decision Letter 2]

12 Aug 2022

Anemia in female patients with myasthenia gravis

PONE-D-22-12752R2

Dear Dr. Suzuki,

We’re pleased to inform you that your manuscript has been judged scientifically suitable for publication and will be formally accepted for publication once it meets all outstanding technical requirements.

Kind regards,

Jong-Ling Fuh

Academic Editor

PLOS ONE
---

## [Editor Report · Acceptance letter]

26 Aug 2022

PONE-D-22-12752R2 

Anemia in female patients with myasthenia gravis 

Dear Dr. Suzuki:

I'm pleased to inform you that your manuscript has been deemed suitable for publication in PLOS ONE. Congratulations! Your manuscript is now with our production department. 

Kind regards, 

on behalf of

Dr. Jong-Ling Fuh 

Academic Editor

PLOS ONE